# Serological and viral prevalence of Oropouche virus (OROV): A systematic review and meta-analysis from 2000–2024 including human, animal, and vector surveillance studies

Emilie Toews[1], Sabah Shaikh[1], Shaila Akter[2], Caseng Zhang[3], Anabel Selemon[1], Rahul K. Arora[1], Niklas Bobrovitz[1,4], Thomas Jaenisch[2,5], Harriet Ware[1‡], Mairead G. Whelan[1‡*]

1 Centre for Health Informatics, Department of Community Health Sciences, Cumming School of Medicine, University of Calgary, Calgary, Alberta, Canada, 2 Heidelberg Institute of Global Health, Heidelberg University Hospital, Heidelberg, Germany, 3 Faculty of Medicine, University of Ottawa, Ottawa, Ontario, Canada, 4 Department of Emergency Medicine, Cumming School of Medicine, University of Calgary, Calgary, Alberta, Canada, 5 Center for Global Health, Colorado School of Public Health, Aurora, Colorado, United States of America

‡ HW and MGW are joint senior authors on this work.
* mairead.whelan@ucalgary.ca

## Abstract

### Background

Oropouche virus (OROV) is an emerging arbovirus primarily transmitted by biting midges and is increasingly recognized as a public health threat in Central and South America. With over 11,000 confirmed cases reported in 2024, a ten-fold increase from the previous year, its transmission dynamics and true burden remain poorly understood due to diagnostic challenges and fragmented surveillance systems.

### Objective

This systematic review and meta-analysis (SRMA) synthesizes OROV prevalence data in humans and summarizes the available data for vectors and animal hosts sampled between 2000 and 2024 to provide updated estimates and identify key surveillance gaps.

### Methods

We systematically searched Web of Science, PubMed, Embase, Medline, and LILACS for OROV seroprevalence and viral prevalence studies in human, insect, and animal populations, published up to September 12, 2024. The review protocol was registered with PROSPERO (CRD42024551000). Studies were extracted in duplicate, and data were meta-analyzed using generalized linear mixed-effects models. Risk of bias was appraised using a modified Joanna Briggs Institute checklist.

which permits unrestricted use, distribution, and reproduction in any medium, provided the original author and source are credited.

**Data availability statement:** Study data are available through the interactive and open-access ArboTracker dashboard and data platform, accessible at https://www.serotracker.com/pathogen/arbovirus/dashboard.

**Funding:** MGW, HW, SS, AS, ET, NB, and RKA report funding for this project from the University of Calgary (Transdisciplinary Connector Grant), Canadian Institutes of Health Research (GLB192241), and the Public Health Agency of Canada (through Canada's COVID-19 Immunity Task Force, 2021-HQ-00056). SA and TJ report funding for this project from European Commission ReCoDID and Contagio grants (EC/825746 and EC/101137283) as well as from the Wellcome Trust (DeZi Network: 316633/Z/24/Z). TJ and SA report funding for this project from the German Research Foundation (grant number 451956976). This manuscript does not necessarily reflect the views of the World Health Organization or any other funder. The funders had no role in study design, data collection and analysis, decision to publish, or preparation of the manuscript.

**Competing interests:** I have read the journal's policy and the authors of this manuscript have the following competing interests: MGW, HW, SS, AS, ET, NB, and RKA report additional separate funding, unrelated to the project, from the Canadian Medical Association Joule Innovation Fund, World Health Organization, the Robert Koch Institute, and the Food and Agriculture Organization of the United Nations. MGW received a travel honorarium from the University of Hong Kong, unrelated to this work. RKA is employed at OpenAI and receives equity compensation as part of the standard compensation package; RKA was also previously a venture fellow at Flagship Pioneering, is currently a minority shareholder of Alethea Medical, and has received funding from the Rhodes Trust and Open Philanthropy. TJ report additional funding, unrelated to this project, from the Bill & Melinda Gates Foundation Serosurveillance Grant (GATES/INV-039656) and the Center for Disease Control Air Quality Contract (CDC/75D30123C17606).

## Results

We included 71 articles reporting serological or viral prevalence of OROV across nine countries. Between 2000–2024, pooled human seroprevalence among individuals with febrile illness or suspected of Oropouche infection was 12.6% [95% CI 5.3-26.9%] across four South American countries and seroprevalence of 1.1% [95% CI 0.5-2.3%] was observed in asymptomatic groups. Viral prevalence among individuals with febrile illness or suspected of Oropouche infection was 1.5% [0.8-3.0%] across seven South American countries and Haiti. Most studies used convenience sampling and RT-PCR or hemagglutination assays. In vector populations, positive OROV prevalence in *Aedes aegypti* and *Culex quinquefasciatus* was reported in two of 18 sources, while 10.0% and 7.5% animal host prevalence was reported in dogs and cattle, respectively. We found high risk of bias in 11.3% of studies in our critical appraisal, with most animal, human, and vector studies falling in the moderate risk of bias range.

## Conclusions

Despite rising numbers of OROV reported cases, prevalence estimates remain limited by sparse surveillance and variable methodology. This review highlights the urgent need for standardized serological assays, community-based studies, and expanded surveillance in animal and vector reservoirs. A One Health approach is essential to monitor OROV transmission and inform regional preparedness efforts.

## Author summary

Oropouche virus is a pathogen carried by biting midges and some other insect vectors that causes fever and flu-like illness in people, and it can also be carried by animals. Cases of Oropouche virus have been rising in Central and South America, with approximately 11,000 confirmed cases in 2024 – a ten-times increase from the previous year. Despite the growing outbreaks, we still know little about how many people and animals have been infected; studies testing at-risk populations for viral infection or the presence of antibodies are valuable tools in filling these knowledge gaps. To address this, we reviewed such studies published from 2000 to 2024 that tested people, animals, and insects for Oropouche virus, and produced estimates of the virus' prevalence in these populations to better understand its spread and detection. We found that in humans with suspected infection in four South American countries, i.e., with fever and related symptoms, 12.6% of them had antibodies for Oropouche – indicating a history of infection. The prevalence of active viral infections in similar populations was 1.5%. In asymptomatic people, the prevalence of antibodies was lower, at 1.1%. We also found small presence of the virus in insect vectors and animals, namely dogs and cattle. Overall, the studies had varying methods especially with

regards to diagnostic tests, and the number of studies is still very limited, especially in animals and insects. Our findings highlight the urgent need for better tools, standardized research methods, and an increase in community surveillance studies across species to better understand Oropouche, prevent future outbreaks, and best respond to the health needs of affected communities.

## Introduction

The Oropouche virus (OROV), a vector-borne arbovirus from the *Peribunyaviridae* family typically transmitted by biting midges, is rapidly gaining prominence as a critical public health threat in Central and South America. As of November 2024, the World Health Organization (WHO) reported over 11,000 confirmed cases, marking a tenfold increase from 2023 [1]. OROV infection causes Oropouche fever, an acute illness characterized by fever, headache, joint pain, and, in severe cases, neurological complications.[2] Emerging evidence from recent outbreaks suggests that OROV may have additional, though not yet confirmed, modes of transmission, such as possible vertical transmission during pregnancy, which has raised concerns about potential congenital outcomes, including microcephaly [3]. Furthermore, early data indicate that OROV's vector range may be expanding both geographically and into other potential carrier species, such as *Culex* mosquitos.[4]

Accurate prevalence estimates are critical for tracking OROV's spread and informing targeted public health interventions. Arboviruses like OROV are frequently under-reported or misdiagnosed due to clinical similarities with other acute febrile illnesses, cross-reactivity in diagnostic tests, and the high proportion of mild or asymptomatic cases that evade detection [5]. Additionally, while there has been evidence of infection via neutralizing antibodies in animal reservoirs like birds and mammals, comprehensive animal surveillance remains limited, complicating efforts to assess zoonotic spillover risks.[6] Serological studies provide key insights into population-level exposure and geographic spread among both human and animal hosts, while viral prevalence studies help distinguish between active outbreaks and endemic circulation. Despite the value of these surveillance estimates, data remain largely fragmented across published literature, underscoring the need for systematic reviews and meta-analyses to identify trends.

Recent evidence indicates wide variation in both human seroprevalence and viral prevalence. One study in Colombia estimated 2% seroprevalence in healthy individuals [7], while another estimated 40% seroprevalence in febrile patients in a high-transmission zone in Brazil [8]. Another recent paper estimated 6.3% average seroprevalence in Latin America, including samples pooled from a range of population types [9]. A viral prevalence study in Colombia showed RT-qPCR identified estimates less than 10% among individuals with fever and related symptoms [7,8,10]. Previous reviews by Romero-Alvarez [11] and Walsh [12] have documented case counts and virus detection locations, but lack synthesized prevalence estimates and updated data in the context of recent 2024 outbreaks. Other recent literature reviews have either focused exclusively on human seroprevalence [13] or employed narrative synthesis without meta-analysis [14], limiting understanding of OROV circulation. In contrast, our review provides a comprehensive synthesis of prevalence data across human, vector, and animal hosts.

In this paper, our objective was to systematically review and meta-analyze OROV prevalence data in humans with an emphasis on evidence from the last decade (2000–2024) to provide an updated and comprehensive synthesis. Additionally, we review the literature on OROV prevalence in animal host reservoirs, which we define in this paper as non-human and/or non-insect populations, offering a preliminary overview of its circulation in OROV's circulation in domestic and wild animals.. To enhance accessibility and utility for researchers and public health professionals, the compiled data and methodology from this review are made available through the interactive and open-access ArboTracker dashboard and data platform [15]. By adopting a One Health approach, this study aims to strengthen public health preparedness and guide surveillance strategies to mitigate OROV's impact in the Americas.

## Methods

The protocol for this SRMA is reported according to the Preferred Reporting Items Systematic review and Meta-Analyses (PRISMA) guideline (File A in S1 Text). We searched five databases (Web of Science, PubMed, Embase, Medline, and LILACS) on September 12, 2024, using comprehensive search terms related to OROV and prevalence estimates, developed with input from a health science librarian (File B in S1 Text). Additional articles were found by screening reference lists from reviews published between 2021 and 2024. The review is registered with PROSPERO as part of a review of arbovirus prevalence studies (CRD42024551000, File C in S1 Text) and is openly accessible on the ArboTracker dashboard website (https://new.serotracker.com/pathogen/arbovirus/dashboard).

References were uploaded into Covidence [16] for de-duplication and screening. Titles and abstracts were independently reviewed by pairs of reviewers, followed by full-text screening of eligible articles. A third reviewer resolved discrepancies. Non-English articles were translated via Google Translate or Microsoft Word's translate function. Cross-sectional, case-control and cohort studies were included if they reported OROV prevalence estimates for humans, animal hosts, or insect populations with a specific end date in a defined geographic location. We included peer-reviewed literature, preprints, grey literature, and media reports without language restrictions. Full inclusion and exclusion criteria can be found in File D in S1 Text.

Included articles were uploaded into an AirTable database for data extraction. For each source, we collected bibliographic details and prevalence estimates, along with information on study design, testing methods, and population characteristics. Articles were excluded if a prevalence estimate or number of persons tested was missing. Where an article or source contains multiple prevalence estimates stratified by time, geography, non-overlapping populations, test type, or gender, we split the article into multiple "studies"—for the purpose of this review, "study" means a distinct estimate treated as separate entries for analysis. Overlapping stratifications were avoided except for when test type, and gender subgroup estimates were available in different populations, timeframes, or geographic locations. Data extraction was performed by one reviewer and independently verified by a second, with any disputes being resolved through discussion. Data from included articles can also be found on the ArboTracker dashboard, alongside seroprevalence studies for six other arboviruses as part of a larger living review [15].

We critically appraised all studies using a modified version of the Joanna Briggs Institute (JBI) checklist for prevalence studies that has previously been used in seroprevalence reviews [17]. To assess risk of bias, 6/8 items on the checklist were determined by an automated decision rule, with the final two items verified by dual manual review (each item is outlined in File E in S1 Text). Each study was assigned a rating of low, moderate, or high risk of bias by the decision rule based on the specific combination of JBI checklist ratings for that study. Human and vector studies were assessed using the same criteria. This method has been validated against overall risk of bias assessments derived manually by two independent reviewers for previously collected seroprevalence studies, showing good agreement with manual review (intra-class correlation 0.77, 95% CI 0.74 to 0.80; n = 2,070 studies) [17].

Study and population characteristics were summarized using descriptive statistics (i.e., counts and percentages). To capture timely evidence from the last decade and provide a current-day snapshot of the virus's circulation, we included only studies that concluded sampling between 2000 and 2024 in the meta-analysis. We meta-analyzed the seroprevalence and viral prevalence of OROV in humans using binomial generalized linear mixed-effects models with log link function and reported point estimates and 95% confidence intervals. Subgroup analyses were conducted, stratified by expected sources of study heterogeneity, including country and population type (febrile patients or suspected cases vs. general population). Heterogeneity was assessed using the $I^2$ statistic to quantify the proportion of variation attributable to true differences rather than chance. To explore possible causes of heterogeneity among study results, we constructed a binomial generalized linear mixed-effects model with log link function. Independent predictors were defined a priori as country, population type, sampling end year, and type of assay (neutralizing vs screening only). Analyses were performed in R [meta and metafor packages, R version 4.4.3].

## Results

### Search results

The database searches identified 3,220 abstracts, and an additional 38 unique abstracts were identified from the reference lists of recent reviews [5,12–14,18] (Fig 1). Following the abstract and title screening phase, we screened 131 full-text articles, ultimately including 71 articles in the review. Human and vector studies were included in 59 and 18 of the 71 publications, respectively. The 71 articles contained a total of 287 unique viral or serological prevalence studies eligible for inclusion (detailed references and information available in Tables A and B in S1 Text). Sixty-one articles sampled in 2000 or later were included in the meta-analysis (Fig 1).

### Characteristics of included studies

Nine Caribbean and South and Central American countries were represented among the studies included in the descriptive analysis, including studies in Bolivia, Brazil, Colombia, Costa Rica, Ecuador, French Guiana, Haiti, Paraguay, and Peru. Most studies sampled human populations (60%), with fewer sampling insect (20%) and animal host (19%) populations.

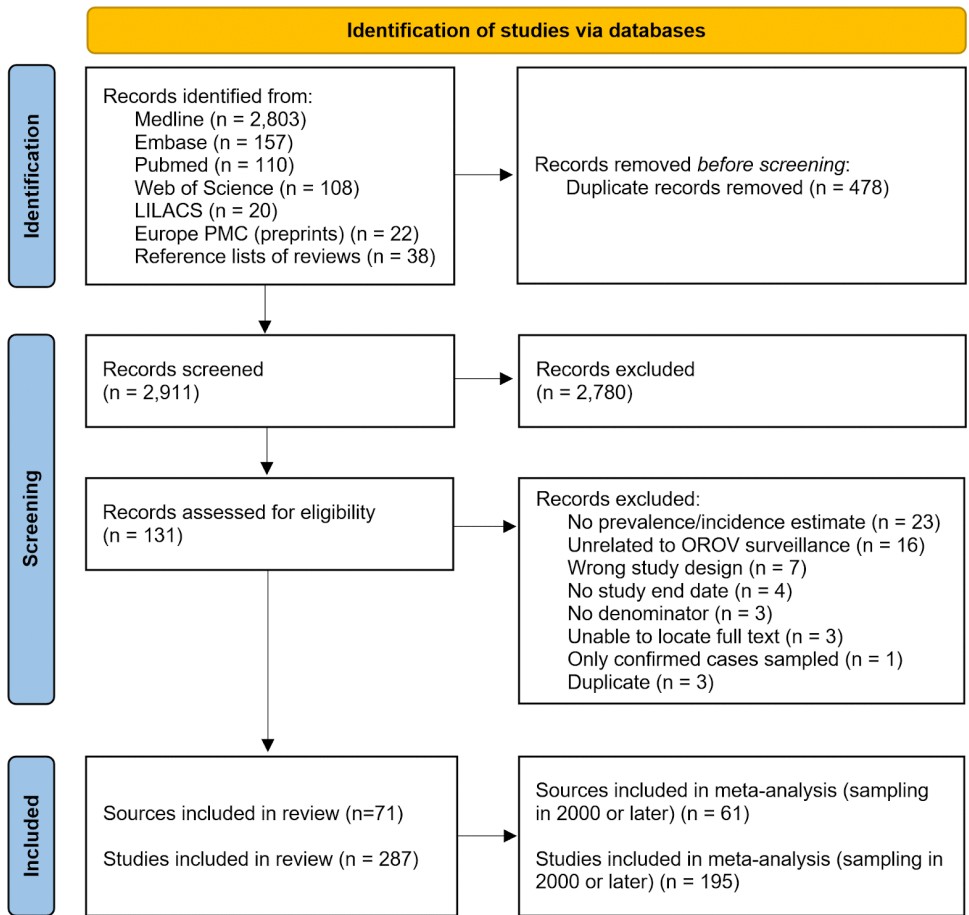

**Fig 1. PRISMA flowchart for inclusion and exclusion of Oropouche virus literature.**

About half of human studies estimated seroprevalence (57%) while 43% estimated viral prevalence (Table 1). Convenience sampling was the most frequent sampling method. Sampling frames in human studies were primarily febrile patients (63%) and asymptomatic communities (27%). In studies sampling in 2000–2024, febrile patients constituted a greater proportion of sampling frames (75%), with a smaller proportion of studies sampling asymptomatic communities (12%). Among the testing strategies used to measure prevalence in humans, most studies used reverse transcription polymerase chain reaction (RT-PCR) (61%) or enzyme-linked immunosorbent (ELISA) (14%) assays. Only studies that concluded sampling in 2000–2024 were included in the subsequent results (n = 195), which included OROV estimates from 99 human studies and 96 vector studies.

### Human seroprevalence

Ninety-two studies reported estimates of human seroprevalence, with 38 studies sampling between 2000 and 2024. Pooled seroprevalence in humans with febrile illness or suspected Oropouche infection between 2000 and 2024 was 12.6% [95% CI 5.3 to 26.9%] (n = 25) with substantial heterogeneity ($I^2$ 99%) (Fig 2). In Brazilian studies (n = 12), seroprevalence ranged from 41.1% to 69.5% in Pará State, which experienced the largest OROV outbreak in the country, and 0.6% to 20.3% outside of Pará State. In Colombia (n = 6), seroprevalence ranged from 5.4% across nationally pooled studies, with higher estimates localized in the Amazonas State at 42.5%. In Peruvian studies (n = 5), seroprevalence ranged from 0.0% across nationally pooled results to 65.4% in Cajamarca State. In Ecuador only two studies were included, which reported two estimates of 0.3% seroprevalence in Pastaza State (Manock et al, 2004).

Pooled seroprevalence among asymptomatic general populations between 2000 and 2024 was 1.1% [95% CI 0.5 to 2.3%] (n = 13) with lower heterogeneity ($I^2$ 92%) (Fig 3). These studies were performed in Brazil (n = 9) and Colombia (n = 4). Pooled seroprevalence was similar in the studies located in these two countries—1.0% [95% CI 0.4 to 2.7%] in Brazil and 1.8% [95% CI 0.7 to 4.7%] in Colombia.

The multivariable meta-regression model of seroprevalence is reported in Table 2. Much of the heterogeneity in effect sizes was explained by country, population type, and sampling end year. By contrast, compared to studies that used neutralizing assays, there were no differences between seroprevalence in studies that used screening assays only (PR 0.854 [0.150 to 4.86], $p = 0.859$).

### Human viral prevalence

Seventy studies reported estimates of human viral prevalence with 61 studies sampling between 2000 and 2024. Pooled viral prevalence among individuals with febrile illness or suspected of Oropouche infection between 2000 and 2024 was 1.5% [95% CI 0.8 to 3.0%] with substantial heterogeneity ($I^2 = 99\%$) (Fig 4). The 61 studies were performed in local studies spanning eight countries, with a range of viral prevalence of 0.2% [95% CI 0.0 to 2.8%] in Bolivian studies, 1.0% [95% CI 0.3 to 3.1%] in Brazilian studies, 1.1% [95% CI 0.3 to 3.8%] in Ecuadorian studies, 3.1% [95% CI 1.3 to 7.0%] in Peruvian studies, and 3.5% [95% CI 0.8 to 14.2%] in Colombian studies. These results include countries where at least three studies were available to synthesize.

### Characteristics of vector studies and vector prevalence

One hundred and twenty-five vector studies reported OROV prevalence, with 96 studies sampling in 2000 or later. Since 2000, OROV prevalence in vector populations has been estimated mostly through RT-PCR (70.8%) rather than through serological methods (29.2%). These studies were derived from insect vector populations (50.0%) and animal vector populations (50.0%) equally. Insect studies were most frequently available for mosquito species (93.8%), although there were 3 studies (6.3%) for OROV prevalence in biting midges (*Culicoides paraensis*) (Table 3). Only four studies reported positive viral prevalence estimates, including in *Aedes aegypti* (1.2% prevalence) [19], *Culex quinquefasciatus* (0.6%; 2.1%) [19,20], and a non-specified mosquito population (0.4%) [21].

PLOS Neglected Tropical Diseases

**Table 1. Human study characteristics (author, year, location, population, sample size etc).**

| Characteristic | All human studies<br>N = 162[1] | Human studies included in meta-analysis (2000 and later)<br>N = 99[1] |
|---|---|---|
| **Country** | | |
| Bolivia (Plurinational State of) | 4 (2.5%) | 4 (4.0%) |
| Brazil | 96 (59%) | 40 (40%) |
| Colombia | 18 (11%) | 18 (18%) |
| Ecuador | 5 (3.1%) | 5 (5.1%) |
| French Guiana | 1 (0.6%) | 1 (1.0%) |
| Haiti | 1 (0.6%) | 1 (1.0%) |
| Paraguay | 1 (0.6%) | 1 (1.0%) |
| Peru | 36 (22%) | 29 (29%) |
| **Estimate Type** | | |
| Seroprevalence | 92 (57%) | 38 (38%) |
| Viral Prevalence | 70 (43%) | 61 (62%) |
| **Sample Frame** | | |
| Asymptomatic Community | 43 (27%) | 12 (12%) |
| Essential non-healthcare workers | 3 (1.9%) | 1 (1.0%) |
| Febrile patients | 102 (63%) | 74 (75%) |
| Positive (PCR) or suspected cases | 10 (6.2%) | 10 (10%) |
| Positive cases of a different arbovirus | 2 (1.2%) | 2 (2.0%) |
| Residual sera | 1 (0.6%) | |
| Students and Daycares | 1 (0.6%) | |
| **Sampling Method** | | |
| Cluster random | 32 (20%) | |
| Convenience | 94 (58%) | 78 (79%) |
| Non-probability sampling | 7 (4.3%) | 3 (3.0%) |
| Simple random | 3 (1.9%) | 3 (3.0%) |
| Stratified random sampling | 6 (3.7%) | |
| Not reported | 20 (12%) | 15 (15%) |
| **Test Type** | | |
| *Screening* | *38 (23.9%)* | *25 (25.2%)* |
| ELISA | 22 (14%) | 14 (14%) |
| MIA | 6 (3.7%) | 6 (6.1%) |
| Other Screening | 10 (6.2%) | 5 (5.1%) |
| *Neutralizing* | *54 (33.1%)* | *13 (13.2%)* |
| HAI | 49 (30%) | 8 (8.1%) |
| PRNT | 5 (3.1%) | 5 (5.1%) |
| *Molecular* | *70 (43.2%)* | *61 (61.1%)* |
| RT-PCR | 60 (37%) | 60 (61%) |
| Viral isolation | 10 (6.2%) | 1 (1.0%) |
| **Assay Target** | | |
| IgG | 7 (4.3%) | 4 (4.0%) |
| IgG,IgM | 6 (3.7%) | 3 (3.0%) |
| IgM | 15 (9.3%) | 13 (13%) |
| IgM,NAb | 6 (3.7%) | 2 (2.0%) |
| M segment (OROV only) | 5 (3.1%) | 5 (5.1%) |

*(Continued)*

**Table 1.** (Continued)

| Characteristic | All human studies | Human studies included in meta-analysis (2000 and later) |
|---|---|---|
| | N = 162[1] | N = 99[1] |
| N Segment | 1 (0.6%) | 1 (1.0%) |
| NAb | 61 (38%) | 14 (14%) |
| S segment (OROV only) | 17 (10%) | 17 (17%) |
| S segment (OROV only),M segment (OROV only) | 1 (0.6%) | 1 (1.0%) |
| Not reported | 43 (27%) | 39 (39%) |
| **Risk of bias** | | |
| Low | 68 (42%) | 47 (47%) |
| Moderate | 63 (39%) | 32 (32%) |
| High | 31 (19%) | 20 (20%) |

[1] n (%)

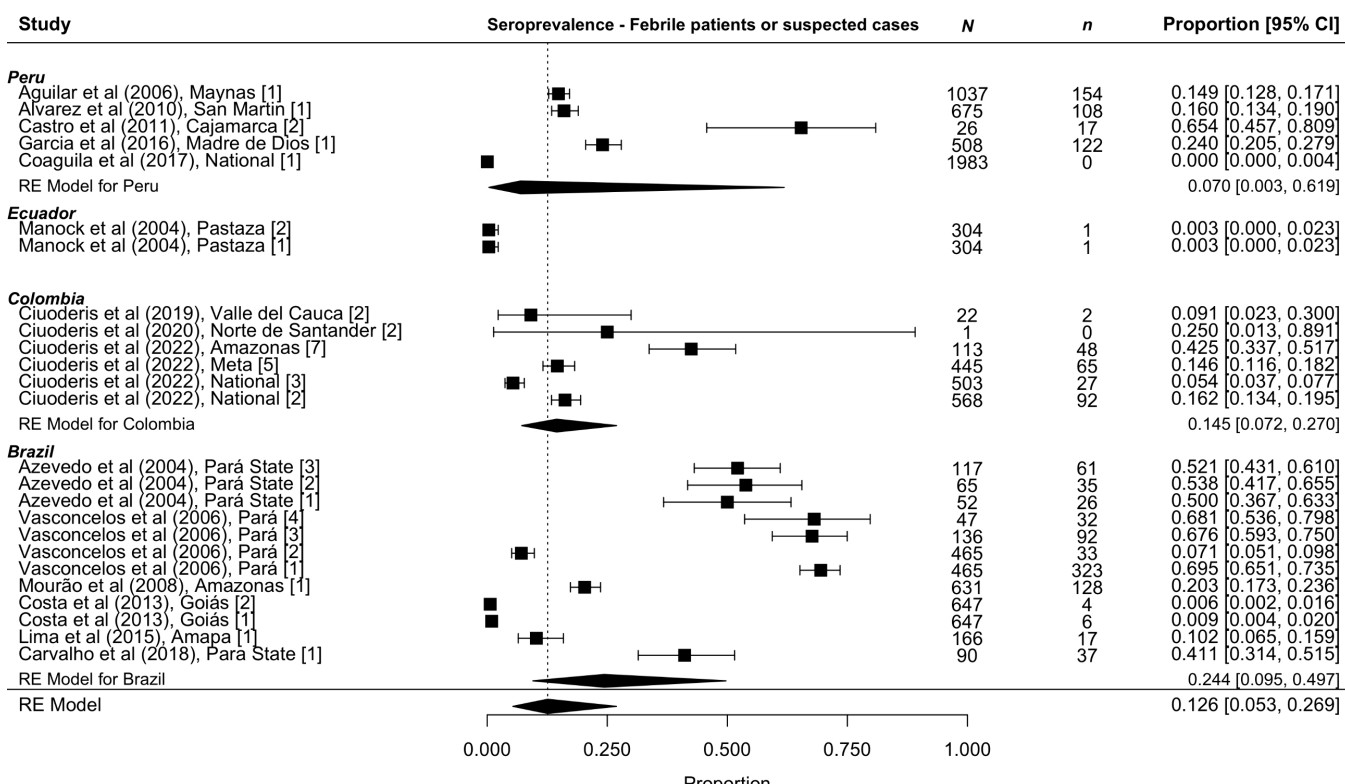

**Fig 2. Reported pooled seroprevalence from studies sampling participants with febrile illness or suspected of Oropouche infection in 2000 to 2024.** Seroprevalence is displayed as a proportion with 95% confidence intervals in square brackets.

For animal vector populations, OROV infection has most frequently been explored in primate species (37.5%), but also in birds (12.5%), opossum (8.3%), horses (8.3%), cows (6.3%), dogs (6.3%), cats (4.2%), coatis (4.2%), sloths (4.2%), and caimans, fish, sheep, and rodents (2.1% each) (Table 3). Evidence of OROV infection was found in seven animal studies sampling after 2000. In three sources sampling four populations of non-human primates with evidence of OROV

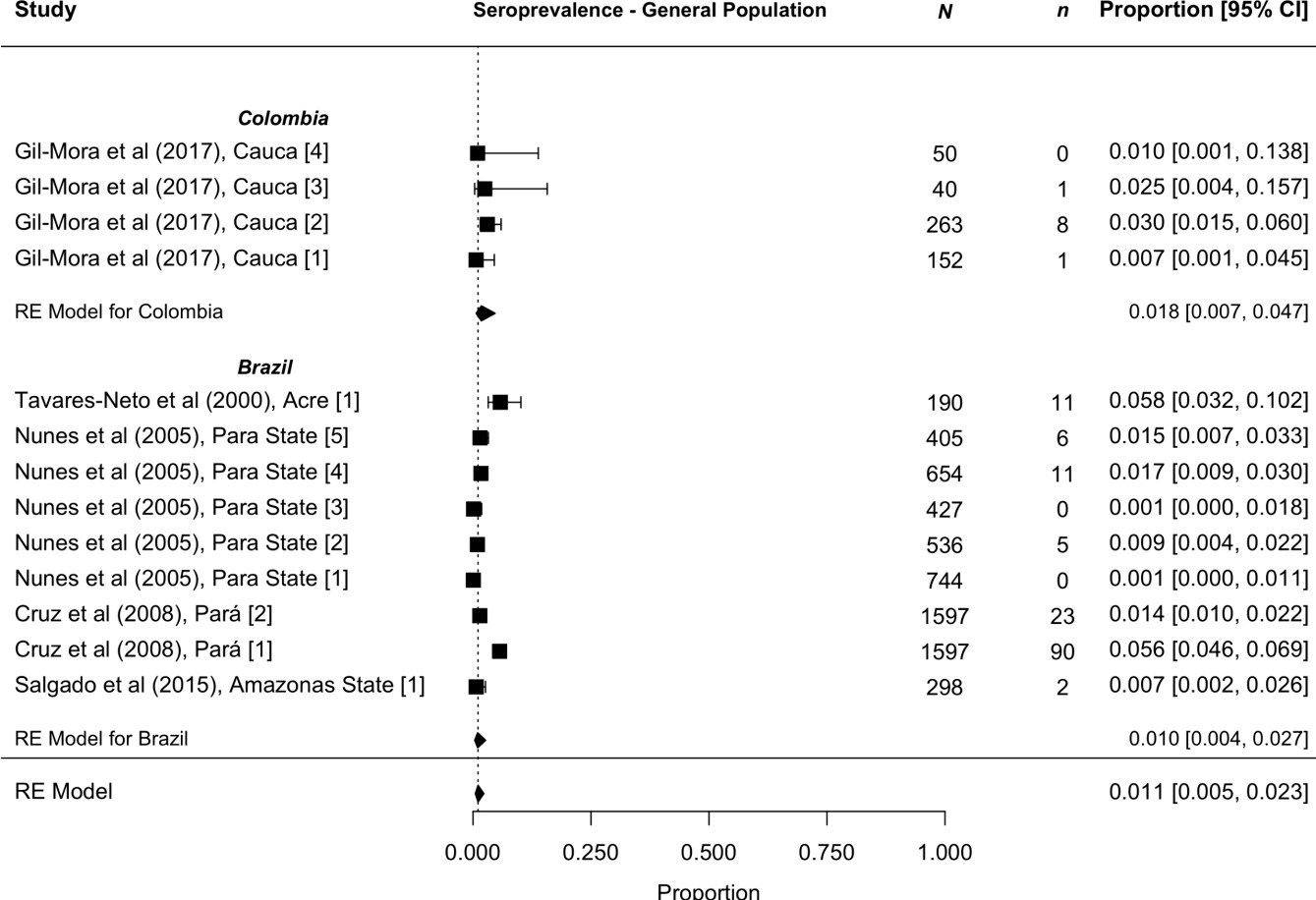

**Fig 3. Reported pooled seroprevalence from studies sampling participants among asymptomatic general populations in 2000 to 2024.** Seroprevalence is displayed as a proportion with 95% confidence intervals in square brackets.

infection, seroprevalence ranged from 1%-7.7% [22–24]. Lastly, two more sources revealed a non-zero OROV seroprevalence in species other than primates, with 10.0% and 7.5% respectively of sampled dogs and cattle [6], and 0.40% of sampled sheep [25] showing past OROV infection (Table B in S1 Text).

## Risk of bias

Most insect (34/48), and animal host(42/50) studies that sampled participants in 2000–2024 were rated moderate risk of bias, while most human (47%; 47/99) studies were rated low risk of bias and only 20% (20/99) of human studies and 8% (8/98) of vector studies had a high risk of bias. Risk of bias was typically higher in vector studies compared with human studies. A summary of the overall risk of bias ratings and a breakdown of each risk of bias indicator for all studies is available in Table C in S1 Text.

## Discussion

### Summary of key findings in humans and context of the current outbreak

This SRMA provides updated prevalence estimates of OROV in humans and vectors in the context of the recent 2024 outbreaks of the virus. Our seroprevalence estimate of OROV in the general population (1.1%) between 2000 and 2024

**Table 2. Meta-regression of seroprevalence from studies sampling participants in 2000 to 2024, n=38.**

| Characteristic | PR | Model 95% CI | p-value |
|---|---|---|---|
| **Country** | | | |
| Brazil | — | — | |
| Colombia | 7.57 | 0.792, 72.40 | 0.079 |
| Ecuador | 0.003 | 0.000, 0.055 | <0.001 |
| Peru | 0.669 | 0.115, 3.89 | 0.655 |
| **Population type** | | | |
| Febrile illness or suspected cases | — | — | |
| General asymptomatic populations | 0.018 | 0.003, 0.110 | <0.001 |
| **Sampling end year** | 0.815 | 0.702, 0.946 | <0.001 |
| **Assay Type** | | | |
| Neutralizing assay | — | — | |
| Screening assay only | 0.854 | 0.150, 4.86 | 0.859 |

PR = Prevalence Ratio, CI = Confidence Interval

The marginal $R^2$, or variation between studies explained only by fixed effects, was 59.6%.

aligns closely with the 1.42% reported by Riccò et al. (2024). Our estimated seroprevalence in febrile individuals (12.6%) was also comparable to the 12.21% reported by Riccò et al. (2024), however it should be noted that convenience sampling in these study populations may impact the estimates. Estimated viral prevalence in febrile individuals (1.5%) across local studies in eight countries in South and Central America was somewhat lower than prior findings (3.86% in Riccò et al. 2024). The difference in viral prevalence may be explained by the difference in sampling periods and the inclusion of high prevalence estimates from the 1970s Brazilian epidemics of OROV in Riccò et al. Given the extensive clinical overlap between arboviral diseases, accurate diagnostic testing and reporting remain critical for effective surveillance and response.

Our findings build upon previous reviews but offer several distinct contributions. While scoping reviews [14,18,26] have examined OROV circulation, none have compared prevalence data across humans, vectors, and animal hosts. Ricco et al. (2024)'s SRMA identified 47 sources compared to 71 in our review, and did not differentiate between sampling time periods, further highlighting the need for updated evidence.

Seroprevalence among the general population was low (1.1%), and the number of studies attempting to estimate OROV burden in asymptomatic or subclinical populations was limited. There was no indication of an increase over time, which could be due to the low number of studies. The higher proportion of studies performed in febrile populations, compared with the general population, underscores the emerging nature of the pathogen, implying that current OROV research and diagnostic efforts are reactionary, where most studies are conducted in response to rising arboviral or febrile disease incidence rather than as part of proactive surveillance efforts. This underscores the urgent need for systematic community-based studies to assess the full scope of OROV circulation, employing a sampling frame where potential geographical spread (and distance from sylvatic regions) are taken into account.

### Key findings in vector and animal reservoir studies

Despite the growing recognition of OROV as a public health threat, research on animal host reservoirs and vector dynamics remains scarce. About half of the studies included in this review examined OROV prevalence in insect or animal populations, with the majority detecting no evidence of current OROV infection. In the vector studies found, mosquito species including *Culex quinquefasciatus* and *Aedes aegypti* were found to harbour OROV [19–21], though it remains unclear

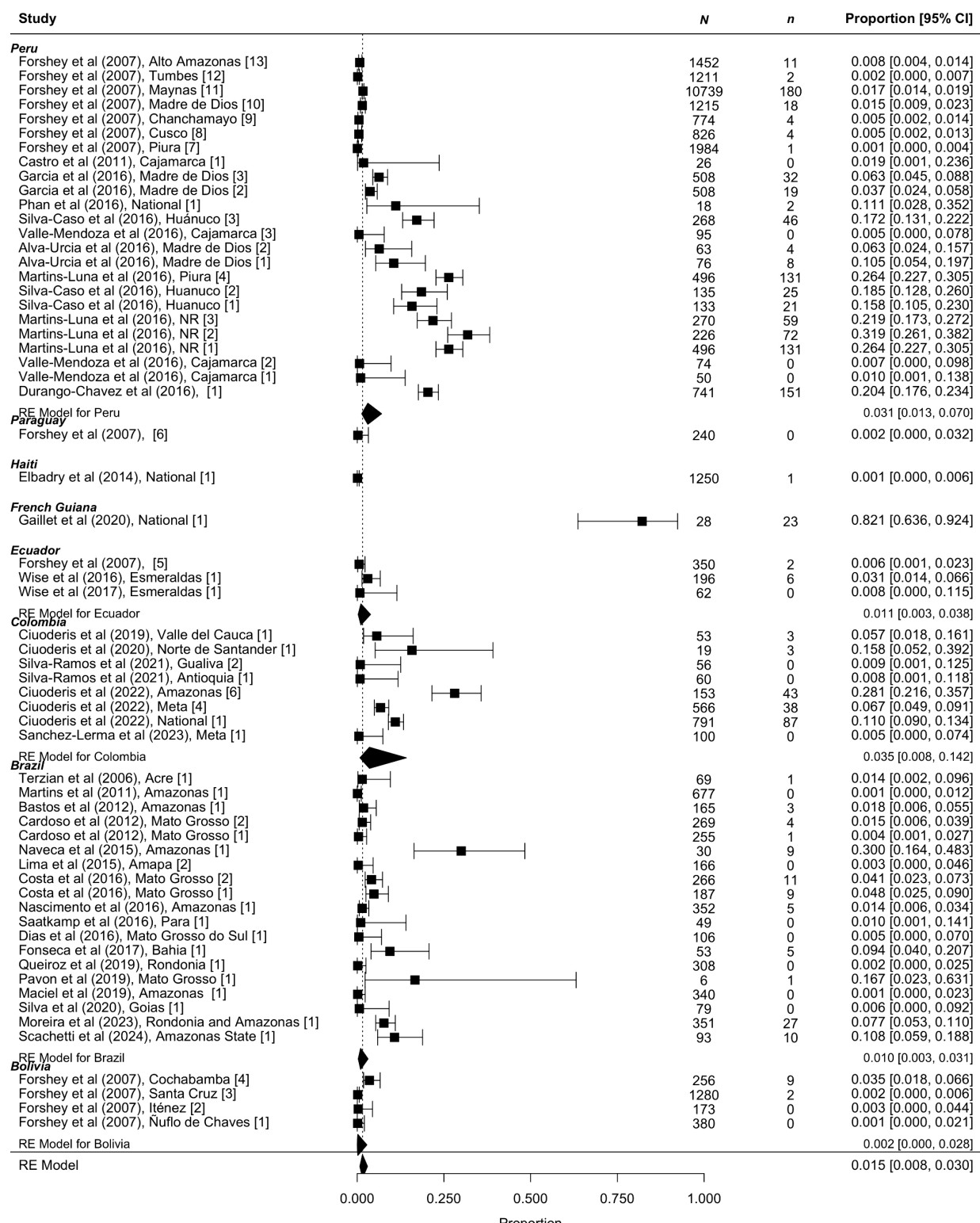

**Fig 4. Reported pooled viral prevalence from studies sampling participants in 2000 to 2024, n=61.** Seroprevalence is displayed as a proportion with 95% confidence intervals in square brackets.

**Table 3. Summary of vector study characteristics.**

| Characteristic | All studies<br>N = 125[1] | Studies included in meta-analysis (2000 and later)<br>N = 96[1] |
|---|---|---|
| **Country** | | |
| Brazil | 122 (98%) | 93 (96.9%) |
| Costa Rica | 2 (1.6%) | 2 (2.1%) |
| Peru | 1 (0.8%) | 1 (1.0%) |
| **Study Population** | | |
| Insect | 71 (57%) | 48 (50%) |
| Animal host | 54 (43%) | 48 (50%) |
| **Estimate Type** | | |
| Seroprevalence | 56 (45%) | 28 (29.2%) |
| Viral Prevalence | 69 (55%) | 68 (70.8%) |
| **Test Type** | | |
| CF | 17 (14%) | |
| ELISA | 1 (0.8%) | 1 (1.0%) |
| HAI | 32 (26%) | 21 (21.9%) |
| PRNT | 6 (4.8%) | 6 (6.3%) |
| RT-PCR | 68 (54%) | 68 (70.8%) |
| Viral isolation | 1 (0.8%) | |
| **Assay Target** | | |
| IgG, IgM | 1 (0.8%) | 1 (1.0%) |
| IgM, NAb | 17 (14%) | |
| NAb | 38 (30%) | 26 (27.1%) |
| NR | 4 (3.2%) | 4 (4.2%) |
| S segment | 65 (52%) | 65 (67.7%) |

[1]n (%)

whether they are capable of sustaining transmission cycles [27]. The detection of OROV in a broad range of vectors is concerning, particularly given the question if *C. quinquefasciatus* and *Culicoides sonorensis* are able to spread the virus beyond its traditional range [28]. Climate change-driven shifts in vector habitats could facilitate OROV transmission in new regions, mirroring the geographic expansion seen in other arboviruses such as West Nile virus [29].

Similarly, the role of animal reservoirs remains poorly understood. Although OROV has been detected in animal hosts such as primates, sheep, dogs, and cattle [6,22–25], the extent to which these species contribute to transmission remains uncertain. Dias et al (2024) detected OROV exposure in 7.5% and 10% of cattle and dogs respectively between 2016 and 2018 but could not find evidence of active infection in the same populations [30]. No studies have sought to quantify OROV prevalence in animal vectors between 2018 and 2024, highlighting a major deficiency in the current body of OROV literature, especially given the current outbreak. Expanding surveillance efforts to include potential wildlife reservoirs is essential for understanding the full epidemiological cycle of OROV and mitigating spillover risks. There were few recent studies of animal and vector populations. Further studies are needed in this area to inform more robust estimates.

Notably, OROV detection in vectors and animal hosts shows both spatial and temporal gaps that complicate the assessment of ongoing transmission. Many vector studies were conducted prior to 2018, with few recent investigations, leaving uncertainty regarding current vector competence and distribution in regions experiencing recent human out-breaks. Geographic overlap between human infections and vector detections—particularly in northern Brazil, Peru, and Colombia—suggests active transmission cycles, yet confirmation of vector competence remains limited to a small number

of studies detecting OROV RNA in non-traditional vectors such as *Culex quinquefasciatus* and *Coquillettidia venezuelensis*. These findings highlight the need for renewed entomological surveillance and comparative studies across vector species to better characterize OROV's ecological range.

### Strengths and limitations

Our study offers a robust and up to date synthesis of OROV prevalence estimates across multiple host populations. Results are available open-access on an interactive ArboTracker dashboard and data platform [15]. This review had limitations. First, significant heterogeneity exists across included studies, stemming from differences in geographic sampling, population demographics, diagnostic assays and outbreak periods. Many studies were locally-scoped and therefore difficult to generalize across larger geographic areas and populations, which is a common challenge when surveillance gaps limit completeness of representative population data. A recent analysis showed that differences in OROV ELISA seroprevalence results between population cohort types (e.g., febrile versus general population) had limited statistical significance when including additional controls for climate variables, which could indicate heterogeneity at least in sample frame is not a substantial limitation, but these results should still be interpreted with appropriate attention [9]. For serological assays in particular, prevalence results are impacted by assay type and performance and whether or not investigators conducted a screening or confirmatory neutralizing test. Cross-reactivity is a known problem for arboviral assays and assay information is poorly reported in published literature, which further limits the interpretation of results [31]. However, use of a valid assay was included as an element in our critical appraisal. Our results suggest that screening and neutralizing tests are comparable in the OROV context, as there was no statistical difference between seroprevalence estimates in studies that used one or the other test (Figs A to D in S1 Text). Future work on the standardization of testing protocols, assay evaluation, and surveillance strategies is needed to enhance comparability across studies. Second, there may be selection bias in the human studies, which primarily used convenience sampling. Particularly in studies in febrile patients, this potential bias could result in over-estimation. As such, these results need to be interpreted with some caution. Ideally, studies would employ random sampling to obtain representative samples of the population. Third, our ability to analyze exposure in animal and vector populations was limited by the small number of studies and cross sectional design of viral testing studies. Further studies are needed in this area.

### Future directions and public health implications

Despite recent advances in OROV research, critical knowledge gaps remain. Community-based seroprevalence studies are needed to assess true population exposure, particularly in non-outbreak periods. Since the completion of our literature search, OROV has been detected in Panama, with imported cases reported in Canada and the Cayman Islands [32]. This geographic expansion further underscores the need for coordinated international surveillance.

Future research should prioritize vector competence studies to clarify the transmission potential of *Culex*, *Aedes*, and other suspected vectors. Standardizing diagnostic assays, including the development of interoperable laboratory protocols, would facilitate cross-study comparisons and improve the reliability of prevalence estimates.

Our findings support the need for comprehensive, standardized arbovirus research study protocols and surveillance programs incorporating OROV. A One Health approach that integrates human, animal, and environmental surveillance will be essential for mitigating future outbreaks, especially given the concerning lack of knowledge on animal and insect hosts in the OROV transmission pathway. In light of the increasing frequency of OROV epidemics, proactive public health interventions, including enhanced vector control and improved diagnostic capacity, should be prioritized.

### Conclusion

This systematic review and meta-analysis provides the most up-to-date synthesis of OROV prevalence estimates in humans, vectors, and potential animal hosts. These findings highlight the increasing public health threat posed by OROV, the gaps in current surveillance efforts, and the need for more systematic studies to inform mitigation strategies.

Addressing these gaps through coordinated research and policy initiatives will be essential for controlling the spread of OROV and reducing its impact on affected populations.

## Supporting information

**S1 Text.** File A: PRISMA Checklist. File B: Search Strategy. File C: PROSPERO Protocol Registration. File D: Full Inclusion and Exclusion Criteria. File E: Risk of Bias Tool Breakdown. Table A: Bibliographic summary of human studies. Table B: Bibliographic summary of vector studies. Table C: Risk of Bias breakdown for all studies. Table D: Summary of species in vector studies. Fig A. Sensitivity Analysis: Reported pooled seroprevalence from studies using screening assays only, sampling participants with febrile illness or suspected of Oropouche infection in 2000–2024. Fig B. Sensitivity Analysis: Reported pooled seroprevalence from studies using neutralizing assays, sampling participants with febrile illness or suspected of Oropouche infection in 2000–2024. Fig C. Sensitivity Analysis: Reported pooled seroprevalence from studies using screening assays only, sampling participants among asymptomatic general populations in 2000–2024. Fig D. Sensitivity Analysis: Reported pooled seroprevalence from studies using neutralizing assays, sampling participants among asymptomatic general populations in 2000–2024.
(DOCX)

**S1 Fig. Renamed c30ef.**
(PNG)

## Acknowledgments

We thank our talented research team and many alumni of the SeroTracker group, as well as the team working at the Universities of Heidelberg and Colorado. We also thank collaborators at the emeritus COVID-19 Immunity Task Force and colleagues affiliated with the Centre for Health Informatics for helpful discussions.

## Author contributions

**Conceptualization:** Emilie Toews, Thomas Jaenisch, Harriet Ware, Mairead G Whelan.

**Data curation:** Emilie Toews, Sabah Shaikh, Shaila Akter, Caseng Zhang, Anabel Selemon, Harriet Ware, Mairead G Whelan.

**Formal analysis:** Emilie Toews, Harriet Ware.

**Funding acquisition:** Rahul K Arora, Niklas Bobrovitz, Thomas Jaenisch, Harriet Ware, Mairead G Whelan.

**Investigation:** Emilie Toews, Sabah Shaikh, Shaila Akter, Caseng Zhang, Anabel Selemon, Thomas Jaenisch, Harriet Ware, Mairead G Whelan.

**Methodology:** Emilie Toews, Sabah Shaikh, Rahul K Arora, Niklas Bobrovitz, Harriet Ware, Mairead G Whelan.

**Project administration:** Rahul K Arora, Niklas Bobrovitz, Thomas Jaenisch, Harriet Ware, Mairead G Whelan.

**Supervision:** Rahul K Arora, Niklas Bobrovitz, Thomas Jaenisch, Harriet Ware, Mairead G Whelan.

**Validation:** Emilie Toews, Mairead G Whelan.

**Visualization:** Emilie Toews.

**Writing – original draft:** Emilie Toews, Sabah Shaikh, Thomas Jaenisch, Harriet Ware, Mairead G Whelan.

**Writing – review & editing:** Emilie Toews, Sabah Shaikh, Shaila Akter, Caseng Zhang, Anabel Selemon, Rahul K Arora, Niklas Bobrovitz, Thomas Jaenisch, Harriet Ware, Mairead G Whelan.

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
