## [Decision Letter · Decision Letter 0]

17 Sep 2025

PNTD-D-25-01092

Serological and viral prevalence of Oropouche virus (OROV): A systematic review and meta-analysis from 2000-2024 including human, animal, and vector surveillance studies

Dear Dr. Whelan,

Thank you for submitting your manuscript to PLOS Neglected Tropical Diseases. After careful consideration, we feel that it has merit but does not fully meet PLOS Neglected Tropical Diseases's publication criteria as it currently stands. Therefore, we invite you to submit a revised version of the manuscript that addresses the points raised during the review process.

Please submit your revised manuscript within 60 days Nov 16 2025 11:59PM. If you will need more time than this to complete your revisions, please reply to this message or contact the journal office at plosntds@plos.org. Please include the following items when submitting your revised manuscript:

We look forward to receiving your revised manuscript.

Kind regards,

Rebecca C. Christofferson

Academic Editor

David Safronetz

Section Editor

Shaden Kamhawi

co-Editor-in-Chief

Paul Brindley

co-Editor-in-Chief

**Journal Requirements:**

4) Thank you for stating "Study data will be made available through the interactive and open-access ArboTracker dashboard and data platform, accessible at We strongly recommend all authors decide on a data sharing plan before acceptance, as the process can be lengthy and hold up publication timelines. Please note that, though access restrictions are acceptable now, your entire data will need to be made freely accessible if your manuscript is accepted for publication. This policy applies to all data except where public deposition would breach compliance with the protocol approved by your research ethics board. 

5) Please ensure that the funders and grant numbers match between the Financial Disclosure field and the Funding Information tab in your submission form. Note that the funders must be provided in the same order in both places as well.

6) Please amend your Competing Interests statement in the online submission form. Please declare all competing interests beginning with the statement "I have read the journal's policy and the authors of this manuscript have the following competing interests:"

Note: If there are no competing interests to declare, please state "The authors have declared that no competing interests exist". 

6) As required by our policy on Data Availability, please ensure your manuscript or supplementary information includes the following:

This information can be included in the main text, supplementary information, or relevant data repository.

**Reviewers' Comments:**

Reviewer's Responses to Questions

**Key Review Criteria Required for Acceptance?**

**Methods**

-Are the objectives of the study clearly articulated with a clear testable hypothesis stated?

-Is the study design appropriate to address the stated objectives?

-Is the population clearly described and appropriate for the hypothesis being tested?

-Is the sample size sufficient to ensure adequate power to address the hypothesis being tested?

-Were correct statistical analysis used to support conclusions?

-Are there concerns about ethical or regulatory requirements being met?

Reviewer #1: The methods section should be described in greater detail. For instance, what type of generalized linear mixed-effects models were used? What distribution and link function were applied? This information is particularly relevant and is currently not adequately reported.

The authors should make a greater effort to identify and understand the sources of heterogeneity among the studies so that appropriate conclusions can be drawn from the results. A meta-regression analysis may be appropriate in this regard.

Reviewer #2: This is not a hypothesis-driven study

I am not concerned about the ethical or regulatory requirements being met.

**Results**

-Does the analysis presented match the analysis plan?

-Are the results clearly and completely presented?

-Are the figures (Tables, Images) of sufficient quality for clarity?

Reviewer #1: The analysis does match the analysis plan.

It is important to specify the within-country locations of the included studies. Generalizing results at the country level may overstate the findings. For example, reporting seroprevalence of 24.4% for Brazil or 14.5% for Colombia may be misleading. The studies reporting the highest seroprevalence in Brazil for instance (Azevedo et al., 2004; Vasconcelos et al., 2006) were conducted in the state of Pará, which accounts for only about 4% of the Brazilian population. Greater effort should be made to account for the within-country origin of the studies and, more importantly, the local characteristics of those areas.

Was any additional analysis performed to investigate the high heterogeneity observed in the results? Based on those values, can the authors reasonably justify grouping these studies together? Probably authors should explore alternative grouping of the results

Reviewer #2: Yes

**Conclusions**

-Are the conclusions supported by the data presented?

-Are the limitations of analysis clearly described?

-Do the authors discuss how these data can be helpful to advance our understanding of the topic under study?

-Is public health relevance addressed?

Reviewer #1: The conclusions should be refined to reflect a more accurate presentation of the results. It does not seem reasonable to use highly localized studies to draw country-wide conclusions.

Only by considering further refinement of results I would confidently state that conclusions are important for the understanding of the topic

Reviewer #2: Yes

**Editorial and Data Presentation Modifications?**

Reviewer #1: Figures: Figure legends should be self-explanatory. For example, clarify what the numbers in brackets represent.

I suggest to authors improve literature support of the statements, specially in the introduction. For example,

First paragraph of introduction: “In addition, early data raise the concern that OROV’s vector range may be expanding both geographically and into other carrying vectors beyond the biting midge, for example Culex mosquitoes” This sentence needs a reference.

Second paragraph of introduction: “Additionally, evidence of infection in animal reservoirs like birds and mammals remains poorly defined, complicating efforts to assess zoonotic spillover risks”. Needs a reference.

Reviewer #2: (No Response)

**Summary and General Comments**

Reviewer #1: This is a relevant study. However, a major concern is the broad generalization of the results. Grouping studies solely by country without considering their local context can be misleading. For instance, stating that febrile individuals in Brazil have an Oropouche seroprevalence of 24.4% is problematic. Closer examination reveals that most of the studies reporting the highest values were conducted in municipalities located in highly sylvatic areas of Brazil (e.g., Amazonas and Pará). Such values should be attributed to those specific regions rather than the entire country, as the current approach may significantly distort the interpretation of the results.

Additionally, the reported heterogeneity is extremely high—above 95%. This issue does not appear to be adequately addressed, especially given the presence of very low estimates (e.g., prevalence between 0.2% and 3.5%). Further analyses are warranted to explain and understand the sources of variability. Possible approaches include meta-regression or more detailed intra-group analyses. It is also possible that important local variables, not considered in the current analysis, could explain part of this variability.

Reviewer #2: This manuscript presents a systematic review and meta-analysis of OROV serological and viral prevalence across human, animal, and vector populations from 2000-2024. The authors identify 71 articles and provide pooled prevalence estimates for humans, as well as a descriptive synthesis for animal and vector data. The review is well-motivated, the methods are largely appropriate, and the open-access data dashboard (ArboTracker) is a valuable resource. The manuscript would benefit from clarifications in data presentation, additional detail in the methods, and a more explicit discussion of certain limitations.

Major Comments

1. The manuscript states that 71 articles were included, but only 61 (human) and 96 (vector) studies were sampled after 2000 for meta-analysis. The text should explicitly state which data were meta-analyzed vs. only described, and why the 2000 cutoff was chosen. Please also clarify how multiple estimates from the same source were handled, e.g., were they pooled within-study or treated as separate entries?

2. The I2 values for the pooled estimates are extremely high (≥ 92-99%). While heterogeneity is acknowledged, the manuscript would benefit from more formal exploration, e.g., meta-regression or sensitivity analysis by test type, sampling frame, or outbreak vs. non-outbreak context.

3. As diagnostic assay performance is a major confounder (particularly cross-reactivity in ELISA/HAI), stratifying or adjusting by assay type could improve interpretability.

4. The limitations section mentions cross-reactivity and poor reporting of assay details, but the impact on pooled prevalence is not quantified. Could you provide a summary table indicating which studies used confirmatory neutralization tests vs. screening assays only? This would allow readers to assess confidence in different subsets of the data.

5. The descriptive synthesis of animal and vector prevalence is valuable but currently feels secondary. The text could be strengthened by summarizing the geographic overlap between human and vector detection. Commenting on the time gap in vector studies, e.g., no studies after 2018 in certain groups, and discussing vector competence evidence in more detail, referencing the limited number of studies showing OROV RNA in non-traditional vectors.

6. The automated decision-rule approach is innovative, but for transparency, the manuscript should provide a brief explanation of the six automated items vs. two manually reviewed items. An example of how different combinations map to low/moderate/high ratings. Clarify if human, vector, and animal studies were assessed using the exact same criteria.

7. Figure 2 and Figure 3: Consider adding the number of studies (n) and total sample size directly in the plot captions or alongside the pooled estimates.

8. Figure 5: Label n and sample size. For all forest plots, displaying raw proportions alongside CIs would make them easier to interpret.

9. Some forest plots (esp. Figure 2) are difficult to read due to density; splitting by country or using facet plots could help.

Minor Comments

1. Introduction “…OROV may additionally be transmitted sexually and…” I may reword this because it is too early to speculate.

2. Line 172: "i.e., with fever and related symptoms" replace “i.e.” with “that is” or rephrase for smoother readability.

3. Ensure consistent terminology: the manuscript alternates between “non-human animals” and “animal hosts”. Pick one and define early.

4. Some tables (e.g., Table 2) contain a lot of detail; consider moving full species lists to supplementary materials for readability.

5. The Discussion could explicitly mention that prevalence estimates in humans may be biased upward in febrile cohorts due to convenience sampling.

6. The conclusion mentions “increasing severity” of OROV epidemics; severity metrics are not presented in this paper, so this should be reworded unless referencing external data.

PLOS authors have the option to publish the peer review history of their article (what does this mean? ). If published, this will include your full peer review and any attached files.

**Do you want your identity to be public for this peer review?** For information about this choice, including consent withdrawal, please see our Privacy Policy .

Reviewer #1: No

Reviewer #2: No

**Figure resubmission:**
---

## [Editor Report · Decision Letter 1]

8 Dec 2025

Dear Ms Whelan,

We are pleased to inform you that your manuscript 'Serological and viral prevalence of Oropouche virus (OROV): A systematic review and meta-analysis from 2000-2024 including human, animal, and vector surveillance studies' has been provisionally accepted for publication in PLOS Neglected Tropical Diseases.

Best regards,

Rebecca C. Christofferson

Academic Editor

David Safronetz

Section Editor

Shaden Kamhawi

co-Editor-in-Chief

Paul Brindley

co-Editor-in-Chief

---

## [Editor Report · Acceptance letter]

Dear Ms Whelan,

We are delighted to inform you that your manuscript, " 

Serological and viral prevalence of Oropouche virus (OROV): A systematic review and meta-analysis from 2000-2024 including human, animal, and vector surveillance studies," has been formally accepted for publication in PLOS Neglected Tropical Diseases.

Best regards,

Shaden Kamhawi

co-Editor-in-Chief

Paul Brindley

co-Editor-in-Chief
